# Zinc Plating on Inkjet-Printed Ti$_3$C$_2$T$_x$ MXene: Effect of Electrolyte and PEG Additive

Prisca Viviani, Eugenio Gibertini * , Vittorio Montanelli and Luca Magagnin

Dipartimento di Chimica, Materiali e Ingegneria Chimica "Giulio Natta", Politecnico di Milano, Via Mancinelli, 7, 20131 Milano, Italy; prisca.viviani@polimi.it (P.V.); vittorio.montanelli@mail.polimi.it (V.M.); luca.magagnin@polimi.it (L.M.)
* Correspondence: eugenio.gibertini@polimi.it

**Abstract:** Zinc-ion batteries (ZIBs) are currently being studied as an alternative to lithium-ion batteries (LIBs). The nucleation and growth of the zinc deposition mechanism is a critical field of research in ZIBs, as it directly affects the battery efficiency and lifespan. It is of paramount importance in mitigating the formation of porous, dendritic Zn structures that may cause cell inefficiency and, eventually, short-circuiting failures. Interfacial engineering plays a key role in providing reversible plating and stripping of metallic Zn in ZIBs through the proper regulation of the electrode–electrolyte interface. In this work, we investigated the behavior and characteristics of Zn plating on inkjet-printed Ti$_3$C$_2$T$_x$ MXene-coated substrates according to the different electrolyte compositions. Specifically, ZnCl$_2$ and ZnSO$_4$ solutions were employed, evaluating the effect of a relatively low-molecular-weight polyethylene glycol (PEG400) addition to the electrolyte as additive. Electrochemical analyses demonstrated higher deposition kinetics in chloride-based electrolytes rather than sulfate ones, resulting in lower nucleation overpotentials. However, the morphology and microstructure of the plated Zn, investigated via scanning electron microscopy (SEM) and X-ray diffraction (XRD), revealed that the electrolytic solution played a predominant role in the Zn crystallite formation rather than the Ti$_3$C$_2$T$_x$ MXene coating. Specifically, the preferential Zn [002] orientation could be favored when using additive-free ZnSO$_4$ solution, and a PEG addition was found to be an efficient texturing agent only in ZnCl$_2$ solution.

**Keywords:** zinc-ion batteries; inkjet printing; Ti$_3$C$_2$T$_x$ MXene; coating; nucleation; overpotential

## 1. Introduction

Environmentally responsible, reliable, and long-lasting energy power sources are nowadays highly demanded and are at the core of research and industry attention. In this frame, electrochemical energy storage systems represent valid solutions as they reliably store energy in case of necessity, being able to compensate for the limitations of renewable energies linked to intermittency and instability. Lithium-ion batteries (LIBs) are nowadays the dominant technology in electric vehicles and portable electronics applications because of their high power and energy densities [1,2]. However, LIB alternatives are currently under study because of the limited availability of lithium and the general high costs associated with the production of these devices. Moreover, safety concerns associated with the use of organic, poisonous electrolytes pose severe limitations on the large-scale adoption of LIBs [3–5]. Among such alternatives, zinc-ion batteries (ZIBs) have attracted interest because of their high element earth-abundance, high Zn specific capacity (820 mAh g$^{-1}$) and lower estimated costs (US \$65 kW h$^{-1}$ for ZIBs vs. over US \$300 kW h$^{-1}$ for LIBs) [6]. In particular, the low Zn redox potential, i.e., −0.763 V vs. SHE, allows ZIBs with aqueous electrolyte chemistry that are intrinsically safer with respect to the organic one used in LIBs [7,8]. However, ZIBs are usually plagued by zinc mossy growth, resulting in poor coulombic efficiency of charge and discharge cycles as a result of the poor electrification of

the loosely bounded Zn mossy structures, eventually leading to battery failures [9–12]. Such phenomenon, which is worsened in alkaline electrolytes, is usually associated with a low surface potential and/or uneven Zn ion diffusion toward the anode electrode surface. In this case, Zn ions are preferentially reduced on Zn protuberances rather than on the flat Zn surface, according to the well-known "tip effect" [13–15]. Considering this, researchers are applying considerable effort toward finding solutions able to erase or mitigate Zn dendritic growth, favoring a homogeneous Zn-plating process. These include the development of novel structural designs of Zn anodes, electrode–electrolyte interface optimization by the formation of in-situ or ex-situ interfacial coatings and protective layers, and electrolyte additives as well as functionalized separators [11,16–18].

Organic and inorganic electrolyte additives can efficiently inhibit Zn dendrite formation, favoring the growth of a smooth and bright surface, usually achieved thanks to the preferential [002] orientation of Zn crystals [19,20]. More specifically, such additives are able to create a blocking barrier for the reactants, i.e., zincates, at the electrode surface, favoring a more even deposition and suppressing preferential Zn growth on dendrites [21]. In particular, among the various organic additives, polyethylene glycols (PEGs) were extensively investigated, whose inhibiting effects derive from the oxygen radicals used for adsorption [22], in which it was shown how their addition to an electrolyte can influence the morphology of the zinc deposit. Specifically, grain sizes were reduced, producing smooth deposits [22,23].

In terms of interfacial engineering, "zincophilic" coatings such as $Ti_3C_2T_x$ MXene (MX) were recently demonstrated to be particularly effective as an interfacial layer to achieve smooth deposition on metal anodes such as Li, Na, K, and Zn [24–27]. $Ti_3C_2T_x$ MXene can unlock fast electrochemical kinetics in the plating/stripping process due to its high electronic conductivity and rapid Zn-ion diffusion. In fact, $Ti_3C_2T_x$ MXene has also been employed in Zn-ion supercapacitors owing to their outstanding Zn-ion adsorption capability [28,29]. Moreover, their hydrophilic functional groups improve electrode wettability [30–33] with electrolytes, resulting in lower interface impedance and more uniform ion flux [34,35]. In particular, the HCP crystal structure of $Ti_3C_2$ MXene was recently demonstrated to be particularly "zincophilic" [24,27,34,36–39] because of the synergistic effect of a matching lattice degree between $Ti_3C_2T_x$ and Zn and because the presence of halogenated terminations on the $Ti_3C_2T_x$ surface favors the production of a coherent heterogeneous MX/Zn interface, which influences and regulates the subsequent homogeneous Zn growth [35,40,41].

In this work, we investigated the effect on the Zn nucleation and growth of both a thin, zincophilic $Ti_3C_2T_x$ substrate and a PEG400 additive in Zn aqueous electrolytes. Inkjet printing was employed as a suitable technique to quickly achieve $Ti_3C_2T_x$ uniform coatings on SS316 substrates. The mechanism of Zn plating in $ZnCl_2$ and $ZnSO_4$ solutions with and without PEG400 was investigated by electrochemical potentiodynamic techniques as well as microstructural and morphological analyses in order to assess the best condition for optimal Zn growth. It was found that the quality of the Zn layer grown on the $Ti_3C_2T_x$ layer, in terms of microstructure, strictly depends on the kind of electrolyte used, and PEG400 was demonstrated to effectively improve the homogeneity of Zn plating only in $ZnCl_2$ electrolytes.

In this work, inkjet printing was employed as a non-conventional technique, rarely reported in the literature, for the surface modification of battery electrodes. A printable $Ti_3C_2T_x$ MXene ink was formulated and jetted onto SS316 substrates, providing uniform $Ti_3C_2T_x$ coatings. We focused our investigations on the effect of the thin $Ti_3C_2T_x$ coating and the electrolyte composition with and without PEG400 additive on the Zn nucleation and growth behavior. Electrochemical potentiodynamic techniques as well as microstructural and morphological analyses assessed the best conditions for optimal Zn growth. It was found that, in terms of microstructure, the quality of Zn grown on the printed $Ti_3C_2T_x$ strictly depends on the kind of electrolyte used, and PEG400 was demonstrated to effectively improve the homogeneity of Zn plating only in $ZnCl_2$ electrolytes.

## 2. Materials and Methods

*Ti$_3$C$_2$T$_x$ MXene Synthesis*—Ti$_3$C$_2$T$_x$ MXene was synthesized through the mixed-acid approach [31,42]. A mass of 10 g of Ti$_3$AlC$_2$ MAX phase (Luoyang Advanced Material Co., Ltd., Luoyang, China) was added to an etchant solution composed of 30 mL of HF, 120 mL of HCl, and 60 mL of H$_2$O. The solution was left to stir at 300 rpm for 24 h at 35 °C. The obtained multilayered Ti$_3$C$_2$T$_x$ was washed multiple times by centrifugation at 3500 rpm for 5 min until a pH of 5–6 was obtained. A final delamination step of the multilayered Ti$_3$C$_2$T$_x$ phase was performed, adding 2 g of Ti$_3$C$_2$T$_x$ to a solution of 40 mL of water and 2 g of LiCl. The solution was placed in a vial and agitated with a mixer homogenizer for 30 min at 2400 rpm. Then the solution was subjected to this series of centrifuges: 3 cycles at 3500 rpm for 10 min, 1 cycle at 5000 rpm for 1 h, and 1 cycle at 8000 rpm for 20 min, discarding the transparent supernatant and redispersing the sediment between each cycle.

*Ti$_3$C$_2$T$_x$ MXene Ink Preparation and Printing*—The Ti$_3$C$_2$T$_x$ ink preparation was re-adapted from a previously reported protocol [43] and prepared according to the following procedure: 40 mL of the delaminated suspension were placed in a beaker, and 0.1 g of Na-asc was added. The solution was stirred for 10 min, transferred to a bath sonicator (20 W/L) for 30 min, and then to a probe sonicator (2 s on, 2 s off, 180 W, 30 min of total time). The Ti$_3$C$_2$T$_x$ dispersion was finally centrifuged for 10 min at 1000 rpm, and ultimately, 3 mM of LDS (lithium dodecyl sulfate) were added to obtain the final ink. A flat-bed thermal inkjet printer (Breva, IJet2L, San Vittore Olona, Italy) equipped with a Ti$_3$C$_2$T$_x$ ink-filled HP45 cartridge was employed for inkjet printing operations. The native printer resolution was 600 dpi. Typically, the substrates were placed on the flat bed and heated at 40 °C, and 20 layers of the Ti$_3$C$_2$T$_x$ ink were consecutively overprinted to achieve a consistent MXene load. After printing, the Ti$_3$C$_2$T$_x$-coated samples were dried in an oven at 70 °C for 1 h.

*Characterization*—electrolytes were prepared by dissolving 2M ZnCl$_2$ or 2M ZnSO$_4$ in water solution with and without 10% wt. PEG400 (Sigma-Aldrich, Saint Louis, MO, USA). Electrolyte conductivity was investigated through a portable multi-range conductometer (Hanna Instruments, Villafranca Padovana, Italy). For electrochemical analyses, 20 layers of the Ti$_3$C$_2$T$_x$ ink were printed on an SS316 disk of 15 mm diameter. Cyclic voltammetries were performed through a Squidstat Plus potentiostat (Admiral Instruments, Tempe, AZ, USA) using a pure Zn wire (diameter 1.5 mm) as the counter electrode and Ag/AgCl (3M KCl) as the reference electrode. All the electrochemical tests were performed without stirring the solution. Tafel plots were built on the cathodic scan only of the 2 mV s$^{-1}$ cyclic voltammetries. Electrochemical impedance spectroscopy (EIS) measurements were performed with the same three-electrode configuration described above using a Biologic VSP-300 potentiostat equipped with an EIS channel in the frequency range 100 kHz–100 mHz using 10 mV as pulse amplitude at open circuit potential. All measurements were performed at room temperature. Hull cell tests were carried out using a small 3D-printed (polypropylene) trapezoidal-shape Hull cell (33 mL volume, long side of 65 mm, short side of 25 mm, height of 32 mm). A 20-printed layer of Ti$_3$C$_2$T$_x$ on a steel plate (34 mm × 50 mm × 1 mm) and a Pt plate (32 mm × 32 mm × 2 mm) were used as the cathode and anode panels, respectively, fixing the current at 100 mA for 20 min.

Zn deposit morphology was investigated through a scanning electron microscope (SEM, Zeiss EVO 50 EP) equipped with an EDX detector (x-sight detector, Oxford instruments, Abingdon, UK), and microstructure was analyzed with an X-ray diffraction (XRD) microscope (model PW1830, Kα1Cu = 1.54058 Å, Philips, Eindhoven, The Netherland). The crystallite size (D) of plated Zn crystals at 5 mA cm$^{-2}$ was calculated by means of the Scherrer equation, D = (0.94 λ)/(β cosθ), where λ is the X-ray wavelength (λ = 1.54058 Å), β is the line broadening at half the maximum intensity (FWHM), and θ is the Bragg angle of the [002] peak. Dynamic light scattering (DLS) was used to investigate the Ti$_3$C$_2$ MXene nanosheet size distribution in the aqueous ink (Zeta sizer Nano, Malvern, UK) at a scattering angle of 173° (backscatter).

## 3. Results

$Ti_3C_2T_x$ MXenes were successfully obtained by the well-known procedure of mixed acid (HF-HCl) etching and subsequent delamination in LiCl solution. Successful etching was confirmed by SEM analyses (Figure S1a), which clearly showed the "open-book" morphology of the particles as a result of the Al selective etching [33]. EDX analyses (Figure S1b,c) also reported a very low average Al-to-Ti atomic ratio (0.041), confirming successful Al removal. On the other hand, XRD (Figure S2) assessed the successful delamination into individual $Ti_3C_2T_x$ MXenes sheets because of the left shift of the intense [002] peak to a very low diffraction angle (6°) as well as the progressive disappearance of the higher-order peaks [44,45]. According to the average hydrodynamic diameter measured by DLS analysis (Figure S3), the average lateral size of the 2D MXene sheets was estimated to be 302 nm [41].

Inkjet printing is a well-known technique capable of quickly producing thin coatings on any substrate by the jetting of small droplets of a colloidal nanoparticle suspension through the nozzles of the printhead. Despite its wide industrial employment and suitability for the interfacial engineering regulation of the electrodes, poor attention was devoted to inkjet printing, and the usual slurry electrode coating is usually preferred. The $Ti_3C_2T_x$ MXene ink we formulated was easily printable, and $Ti_3C_2T_x$-coated electrodes could be quickly produced. The consistency and repeatability of the printing process were checked by weighting the samples. Figure S4 shows that even after more than 1 month of ink storage, the printed $Ti_3C_2T_x$ mass was highly reproducible, meaning that the as-prepared ink was stable and no agglomeration or nozzle clogging occurred. Samples were prepared by printing 20 consecutive overlayers, resulting in a $Ti_3C_2T_x$ mass load of 0.452 mg cm$^{-2}$. The EDX mapping (Figure S5) of the bright bluish $Ti_3C_2T_x$-coated electrodes revealed a very smooth surface and uniform elemental distribution. The thickness of the dried $Ti_3C_2T_x$ 20-layer coating was estimated to be ~2 μm by the SEM analysis of the cross-section (Figure S6).

The CVs of the $Ti_3C_2T_x$-coated electrodes in chloride or sulfate-based zinc electrolytes, with and without 10% wt. PEG400 as additive, are reported in Figure 1a. At a low scan rate (2 mV s$^{-1}$), the typical Zn plating and stripping behavior was clearly visible for each of the investigated electrolytes, with a sloped cathodic curve and a broad anodic peak corresponding to the Zn reduction (plating) and oxidation (stripping), respectively. As reported in Table 1, in terms of charges and current density peaks, the chloride-based electrolyte generally resulted in the highest values when compared to the sulfate-based one.

For instance, the charge related to Zn plating ($Q_{plat}$) in the PEG-free 2M $ZnCl_2$ electrolyte was 3.34 higher than the PEG-free 2M $ZnSO_4$. Evidently, PEG addition induced a reduction of the electrochemical activity, suggesting that the kinetics of the Zn plating were partially hindered by the PEG400 macromolecules. In fact, when the $Q_{plat}$ is considered, a 10% wt. PEG400 addition resulted in a 36.8% reduction in $ZnCl_2$-based electrolytes and 55.4% in the $ZnSO_4$-based one. The higher extent of the zinc ion reduction and consecutive zinc metal oxidation, with respect to the electrolyte composition, can be related to faster nucleation and growth rate. In particular, the nucleation process strongly depends on the nucleation overpotential ($\eta$), as defined in the following equation [46,47]:

$$\eta = \eta_{onset} - \eta_{cross} \tag{1}$$

where $\eta_{cross}$ is the crossover potential corresponding to the equilibrium growth regime, and $\eta_{onset}$ is the onset potential of the reduction process [46]. Both $\eta_{onset}$ and $\eta_{cross}$ were extracted by cyclic voltammetries, as shown in Figure 1b. Values of the nucleation overpotential are reported in Table 1, and $\eta$ coherently increased when shifting from $ZnCl_2$ to $ZnSO_4$-based electrolytes as well as when adding PEG400. More specifically, the $\eta$ values increased to different extents when PEG400 was added to the $ZnCl_2$ or $ZnSO_4$ solutions. In 2M $ZnCl_2$ + 10 wt.% PEG400, the overpotential value increased just slightly from 62 mV to 71 mV, while in the case of 2M $ZnSO_4$ + 10 wt.% PEG400, it consistently rose from 80 mV to 121 mV. This effect can be explained considering the different Zn-ion oligomer formations with PEG400 molecules. In fact, previous work demonstrated that the $Zn^{2+}$-

PEG oligomer strength strictly depends on the counter-ions because of the different Zn ion solvation environments in zinc sulfate solutions with respect to zinc halide (Cl, I, Br) ones [48]. Contrary to zinc halide electrolytes, in $ZnSO_4$ solution, zinc ions are strongly shielded by $H_2O$ molecules that firmly bind to PEG through H-bondings, resulting in stronger Zn-ion oligomer formation and, as a consequence, an overpotential increase. The nucleation overpotential is known to have an impact on the Zn nuclei radius, which is given by the following equation [14,46]:

$$r_n = 2\frac{\gamma V_m}{F|\eta|} \tag{2}$$

where $\gamma$ is the surface energy of the interface between Zn and electrolyte, $V_m$ is the molar volume of Zn, and $F$ is the Faraday constant. As a consequence, a higher $\eta$ should result in a higher density of smaller nuclei. That was demonstrated to be a favorable condition for dendrite-free growth of Zn. Moreover, the variation of $\eta$ suggested that the nucleation process on the $Ti_3C_2T_x$ surface was strictly affected by the electrolyte composition. It is noteworthy to recall that nucleation and growth processes are affected both by the rate of charge transfer and mass transport, and the substrate mostly plays a role in the nucleation through the surface energy regulation in Equation (2). Mass transport is instead ruled by the ionic conductivity of the electrolyte, and, as reported in Table 2, the ionic conductivity decreased 2-fold from 2M $ZnCl_2$ to 2M $ZnSO_4$, and it further dropped when PEG400 was included in the electrolyte. The ionic conductivity reduction in PEG400 was reasonable considering the increase in solution viscosity as well as the complexing ability of PEG macromolecules with $Zn^{2+}$, forming bulky zinc-ion oligomers [48].

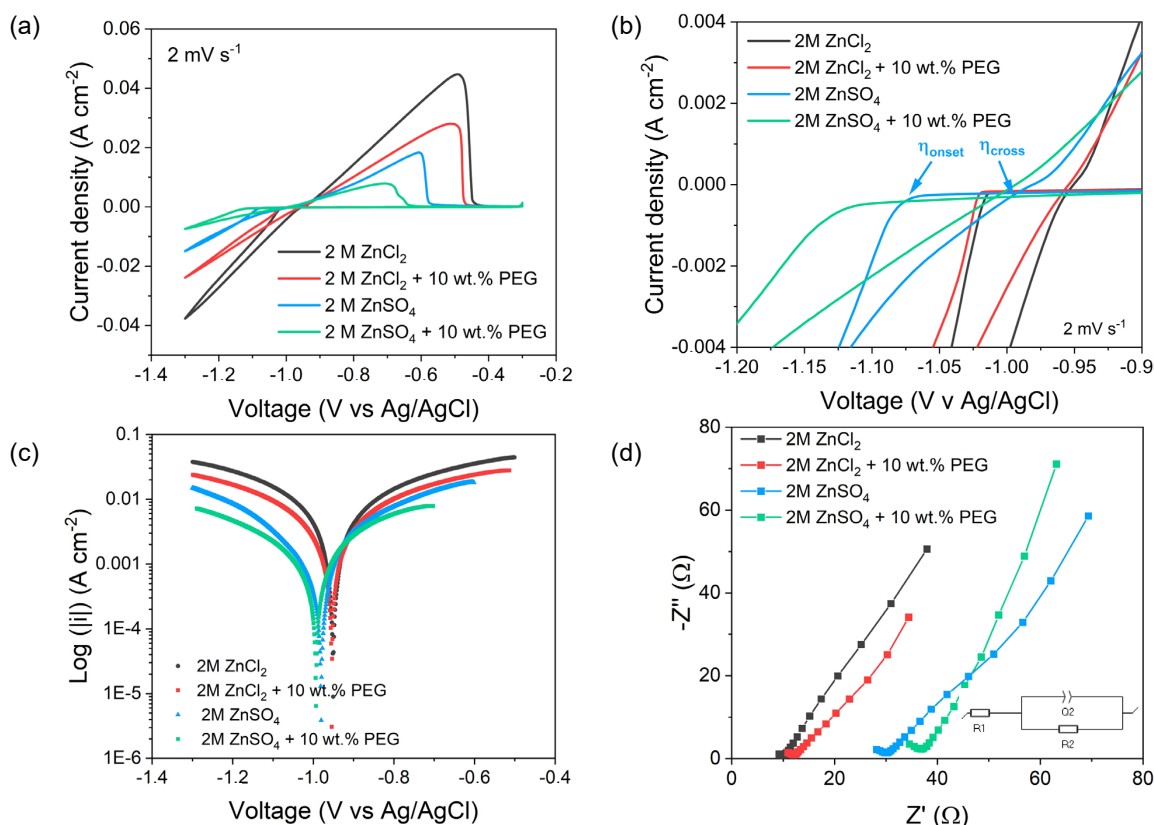

**Figure 1.** (**a**) Cyclic voltammetries of $Ti_3C_2T_x$-coated electrode in different electrolytes recorded at a scan rate of 2 mV s$^{-1}$, (**b**) magnified part of the cyclic voltammetries, highlighting the cross potential and onset potential (an example is pointed out for the 2M $ZnSO_4$ solution), (**c**) Tafel plots obtained from the cathodic scan of CVs in (**a**), (**d**) Nyquist plots of the $Ti_3C_2T_x$ electrodes in different electrolytes (inset: equivalent circuit used for the fitting).

**Table 1.** Electrochemical parameters extracted by cyclic voltammetries at 2 mV s$^{-1}$ according to different electrolytes, i.e., 2M ZnCl$_2$, 2M ZnCl$_2$ + 10 wt.% PEG400, 2M ZnSO$_4$ and 2M ZnSO$_4$ + 10 wt.% PEG400.

|  | 2M ZnCl$_2$ | 2M ZnCl$_2$ + 10 wt.% PEG400 | 2M ZnSO$_4$ | 2M ZnSO$_4$ + 10 wt.% PEG400 |
|---|---|---|---|---|
| \|i$_{p,plat}$\| (A cm$^{-2}$) | 0.037 | 0.024 | 0.014 | 0.007 |
| Q$_{plat}$ (C) | 5.96 | 3.70 | 1.81 | 0.84 |
| I$_{p,strip}$ (A cm$^{-2}$) | 0.044 | 0.027 | 0.018 | 0.008 |
| Q$_{strip}$ (C) | 5.86 | 3.70 | 1.75 | 0.78 |
| η (mV) | 62 | 71 | 80 | 121 |
| i$_0$ (mA cm$^{-2}$) | 10.94 | 6.66 | 2.95 | 1.47 |

**Table 2.** Ionic conductivities of the different electrolytes, as measured by a conductivity meter probe at room temperature.

|  | 2M ZnCl$_2$ | 2M ZnCl$_2$ + 10 wt.% PEG400 | 2M ZnSO$_4$ | 2M ZnSO$_4$ + 10 wt.% PEG400 |
|---|---|---|---|---|
| σ (mS cm$^{-1}$) | 100.2 | 68.1 | 54 | 29.1 |

Considering the cyclic voltammetries in a wide scan-rate range (2–100 mV s$^{-1}$, Figure S7), it is evident that the mass transport limitation regime strictly depended on the electrolyte composition and its ionic conductivity. In fact, at all scan rates, the reduction curves in chloride-based solutions remained sloped, as is typical of a charge-transfer kinetically controlled process. Contrarily, in zinc sulfate-based electrolytes, mass transfer limitations to the reduction process appeared, starting from 25 mV s$^{-1}$ and 10 mV s$^{-1}$ for the ZnSO$_4$ electrolytes with and without PEG, respectively, as the transition from a sloped branch to a well-defined cathodic peak. Comparing the ionic conductivity and nucleation overpotentials, a coherent trend could be pointed out. The exchange current density (i$_0$) values reported in Table 1, obtained by the fitting of linear branches in Tafel plots (Figure 1c), showed that the highest i$_0$ was reported for the ZnCl$_2$ electrolyte, representative of a faster intrinsic rate of the redox reaction at the equilibrium. In PEG-free zinc sulfate electrolyte, the i$_0$ decreased to 2.95 mA cm$^{-2}$ and i$_0$ values lowered by the addition of PEG400, both in ZnCl$_2$ and ZnSO$_4$ electrolytes, consistent with the hypothesis of the Zn-oligomer adsorption, or more generally speaking of the PEG400 macromolecules, which can increase the de-solvation energy of zinc ions [48]. Nyquist plots (Figure 1d) clearly showed a right shift on the Z′ axis with respect to the electrolyte composition as well as an increase in charge transfer resistance. In particular, the electrolyte ionic conductivity was related to the R$_1$ value, while the charge transfer resistance was the diameter of the semicircle and was noted as R$_2$. Both R$_1$ and R$_2$ values are reported in Table 3.

**Table 3.** R$_1$ and R$_2$ values obtained from fitting the semicircle in Nyquist plots with the equivalent circuit reported in Figure 1d. The R$_{2, Eq.3}$ is the charge transfer resistance calculated using Equation (3) and i$_0$ values reported in Table 1.

|  | 2M ZnCl$_2$ | 2M ZnCl$_2$ + 10 wt.% PEG400 | 2M ZnSO$_4$ | 2M ZnSO$_4$ + 10 wt.% PEG400 |
|---|---|---|---|---|
| R$_1$ | 4.98 | 7.2 | 15.71 | 23.27 |
| R$_2$ | 5.55 | 5.85 | 10.55 | 14.94 |
| R$_{2, Eq.3}$ | 4.42 | 7.33 | 10.69 | 16.45 |

It can be easily demonstrated that R$_1$ is representative of the ohmic resistance of the electrolyte since the ratio of ionic conductivities with respect to the 2M ZnCl$_2$ solution corresponded to the ratio of R$_1$ values with respect to the one of 2M ZnCl$_2$ ($\sigma/\sigma_{ZnCl_2} = R_1/R_{1, ZnCl_2}$). On the other hand, the charge-transfer resistance (R$_2$) depends on the energy of de-solvation of ions from the surface, and it is directly related to the rate of the redox

reaction and nucleation overpotential, according to the equation (derived by the Butler–Volmer equation) [14,49]:

$$i = i_0 \frac{zF}{RT}\eta, \quad R_{ct} = \frac{RT}{zFi_0} \tag{3}$$

where $\eta$ is the applied overpotential, $T$ is the temperature, $R$ is the gas constant (8.314 J mol$^{-1}$ K$^{-1}$), $F$ is the Faraday constant (96,485 A s mol$^{-1}$), $z$ is the electron moles, and $i_0$ is the exchange current density. The R$_{2, Eq.3}$, calculated from Equation (3), and the $i_0$ values reported in Table 1 were in good agreement with the R$_2$ obtained by the fitting of Nyquist plots. In summary, both R$_2$ and $i_0$ reflected the fact that in chloride-based solutions, the fast rate of charge transfer, i.e., the fast rate of reversible Zn$^{2+}$ de-solvation and adsorption to the Ti$_3$C$_2$T$_x$ surface, resulted in the lowest nucleation overpotential values, different from the zinc sulfate electrolytes. As a consequence, it was possible that the anions (Cl$^-$ or SO$_4{}^{2-}$) were adsorbed differently on the Ti$_3$C$_2$T$_x$ surface, affecting the charge transfer resistance, as already demonstrated for other species as saccharin anions [47]. Reasonably, PEG400 always increased the R$_2$ for the aforementioned mechanisms [48,50,51].

The electrochemical results were correlated to the morphological and microstructural properties by SEM and XRD analyses on samples prepared by Hull cell. Hull cell is a useful tool widely employed in the galvanic plating industry to reproduce the cathodic plating process. The trapezoidal shape of the Hull cell was designed to allow the investigation of the electroplating characteristic over a wide range of current densities in a single experiment. A picture of the employed Hull cell setup is reported in Figure S8. Figure 2 reports the SEM images of the Zn deposits plated on the Ti$_3$C$_2$T$_x$-coated electrodes according to the specific electrolytes and local current density. Compared to the pristine Ti$_3$C$_2$T$_x$-coated surface (Figure S9), Zn crystals always appeared at both low (1 mA cm$^{-2}$) and higher (5 mA cm$^{-2}$) current densities after 20 min of plating. PEG-free ZnCl$_2$ solutions (Figure 2a–c) produced a compact Zn deposition without a well-defined surface texture and irregular crystal morphology. Moreover, Zn crystal density was evidently higher in the case of 5 mA cm$^{-2}$, as a result of the higher current density, which produced a superior nuclei density. PEG-free ZnSO$_4$ electrolyte (Figure 2g,h) clearly improved the preferential orientation of the zinc HCP crystals, resulting in a more defined morphology of hexagonal-like Zn particles, particularly evident at 5 mA cm$^{-2}$ (Figure 2i). Individual Zn crystals seemed to grow vertically from the Ti$_3$C$_2$T$_x$ substrate rather than forming a very compact and dense layer. A few perfectly hexagonal Zn crystals were evidently growing from the surface of the MXene coating (Figure 2i), with an approximate size of 100 μm. Comparing the PEG-free electrolytes, the electrolyte itself (specifically the counter-ions Cl$^-$ or SO$_4{}^{2-}$) seemed to play a major role in the growth of textured Zn rather than the Ti$_3$C$_2$T$_x$ layer itself. In fact, the Ti$_3$C$_2$T$_x$ coating should promote epitaxial-like growth of the plated Zn layer because of the good matching between the two crystal lattices [40]. However, SEM analyses supported the electrochemical results, suggesting that ion mobility and/or a different anion adsorption at the electrolyte/Ti$_3$C$_2$T$_x$ interface mostly ruled the surface overpotential and the resulting Zn electro crystallization. Increasing surface overpotential was, in fact, demonstrated to be the key feature enabling the smooth growth of compact Zn. Specifically, higher surface overpotential (or higher current density) produces more spatially uniform current distribution and faster growth rate of the [002] planes with respect to the [001] one [14,46,52–54]. Actually, the higher nucleation overpotential and charge-transfer resistance in ZnSO$_4$ electrolytes, contrary to the fast deposition kinetic in ZnCl$_2$ resulting from the electrochemical analyses, were coherent with this view and the actual plated-Zn morphology. A 10% wt. PEG400 addition to the electrolyte greatly affected the plated-Zn morphology but, in the opposite way, when added to the ZnCl$_2$ or ZnSO$_4$ solutions. In ZnCl$_2$ + 10% wt. PEG400, the coating homogeneity was macroscopically greatly improved compared to the PEG-free chloride-based electrolyte. In fact, polyethylene glycols are known to act as a leveling agent; PEG macromolecules adsorbed at the Ti$_3$C$_2$T$_x$ surface hinder the lateral 2D diffusion of zinc ions and the restricted surface migration, resulting in a higher density of small nuclei and more homogeneous Zn. However, SEM

images (Figure 2d–f) didn't show any evident difference at the microscopic scale with respect to PEG-free ZnCl₂ in the surface morphology of Zn grains. Oppositely, in ZnSO₄ + 10% wt. PEG400 (Figure 2g–j), lamellar structures formed, as typical of zinc hydroxide byproducts (e.g., zinc hydroxide sulfate). Zn crystals were scarcely visible and appeared in the whole current density range as irregular, small particles with undefined morphology beneath and within the zinc hydroxide lamellar network. Specifically, the lamellas were more clearly visible at 5 mA cm⁻², suggesting that the ZnSO₄ + 10% wt. PEG400 electrolyte was characterized by a low Zn deposition faradaic efficiency. The local pH rise due to the side hydrogen evolution reaction likely induced the precipitation of the zinc hydroxide structures. The highest charge transfer resistance and limited ionic conductivities detected for the ZnSO₄ + 10% wt. PEG400 solution corroborated this hypothesis [50].

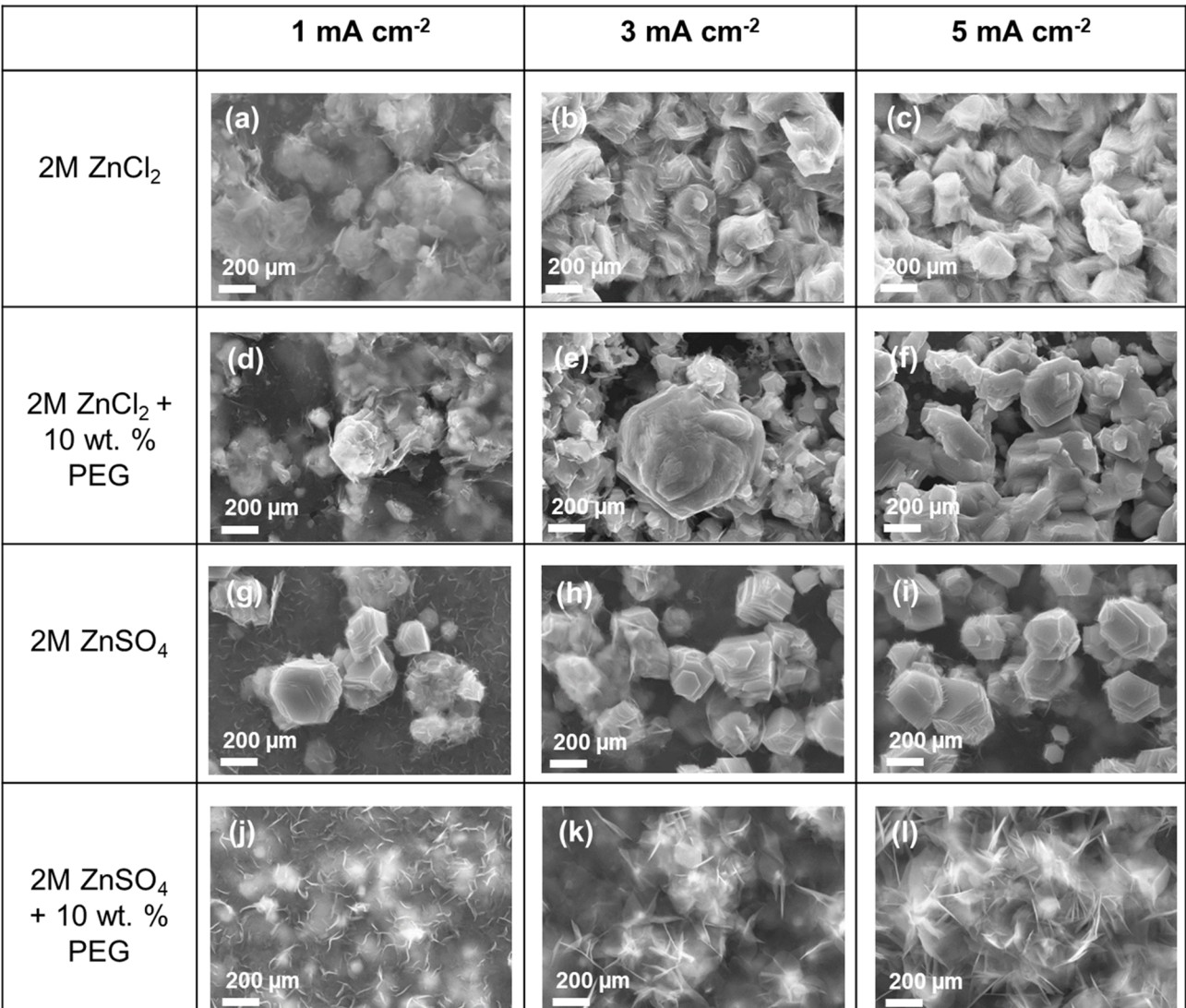

**Figure 2.** SEM images (25 KX) of the Zn structures grown on $Ti_3C_2T_x$ substrate by Hull cell preparation in: (**a**–**c**) 2M ZnCl₂, (**d**–**f**) 2M ZnCl₂ + 10 wt.% PEG400, (**g**–**i**) 2M ZnSO₄ and (**j**–**l**) 2M ZnSO₄ + 10 wt.% PEG400 electrolytes. The corresponding current density value is indicated at the top of each column.

XRD analyses, reported in Figure 3, showed the highest intensity [002] peak associated with the $Ti_3C_2T_x$ MXene coating at 7.5°, while the diffraction peaks of metallic Zn were centered at 36.2°, 38.9°, and 43.2°, corresponding to the [002], [001], and [101] planes, respectively (JCPDS card # 03-065-5973). Insets show the magnified portion of the XRD patterns, where the significant Zn peaks are present. The [002] preferential orientation is

widely recognized to be the most favorable to achieve a low-porosity, dendrite-free metallic Zn coating for highly-reversible plating and stripping in Zn-ion battery applications [53–55]. The ratio between the intensity of the [002] peak and the [101] one ($I_{002}/I_{101}$) was considered to quantify the preferential surface texture [53,54]. In general, the $I_{002}/I_{101}$ increased when moving from a chloride-based to a sulfate-based electrolyte, i.e., 0.49 vs. 1.5, respectively, indicating how the [002] orientation growth was favored, which was compatible with the results obtained by SEM images, where apparently hexagonal-like Zn crystals were obtained with the 2M ZnSO$_4$ electrolyte. Moreover, PEG400 addition induced a slight increase in the $I_{002}/I_{101}$ ratio in both chloride and sulfate-based solutions, i.e., 0.87 and 2.3, respectively. This result suggested that the increased surface overpotential produced by PEG400 effectively improved preferential [002] direction growth, which, however, was not evidently clear from the SEM results. However, as predicted from the SEM analyses, 2M ZnSO$_4$ + 10 wt.% PEG400 electrolyte XRD patterns (Figure 3d) confirmed the presence of small peaks located at ~12°, associated with the formation of Zn hydroxide sulfate structures (JCPDS #39-0690). In terms of crystallite dimensions, ZnCl$_2$ with and without PEG400 and 2M ZnSO$_4$ gave similar results, corresponding to average crystallite sizes of 48.8 nm, 57 nm, and 48.8 nm, respectively. Only in the case of 2M ZnSO$_4$ + 10 wt.% PEG400 was the calculated crystallite size sensibly lower, counting to 37.92 nm coherently with the low efficiency of Zn deposition and lower grain growth rate.

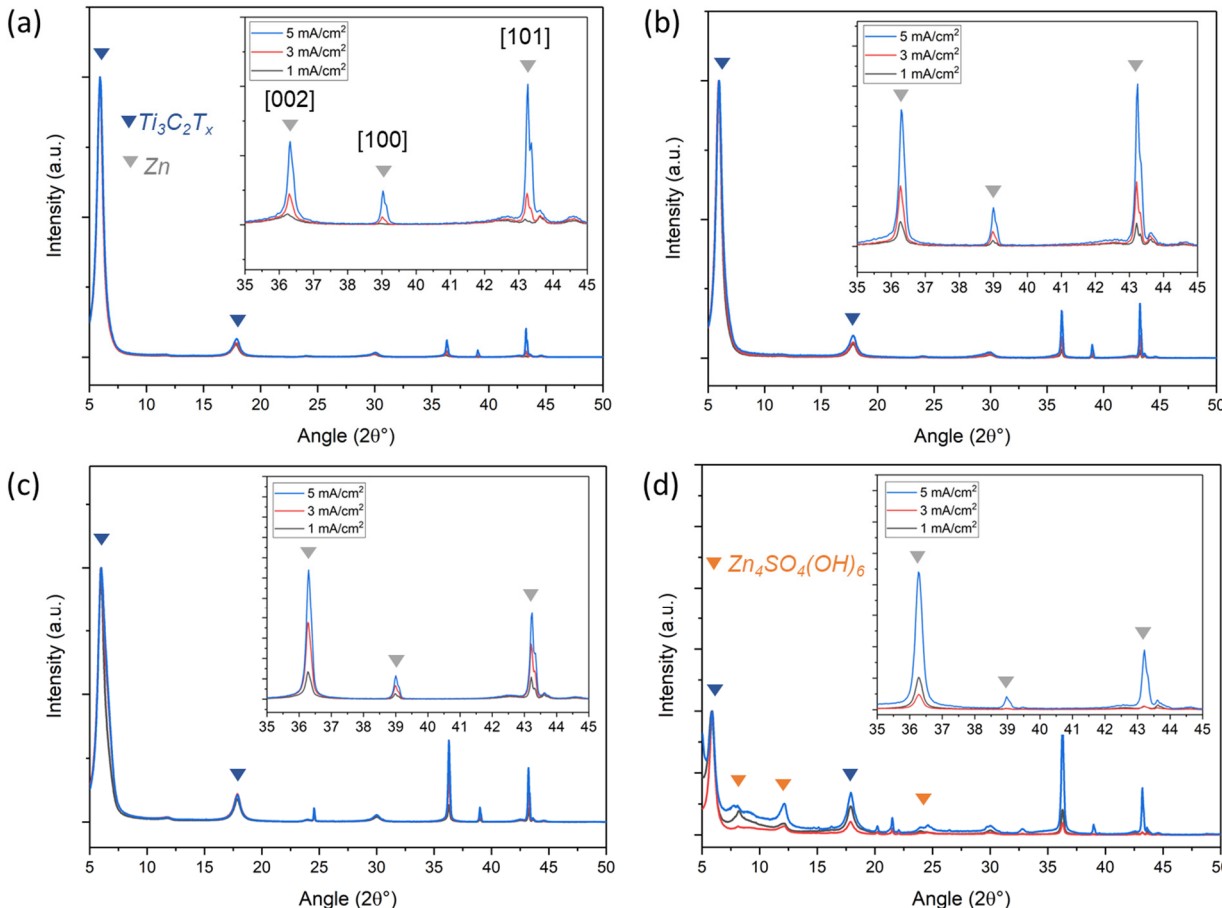

**Figure 3.** XRD diffractograms of Zn deposits obtained by Hull cell sample preparation in different electrolytes: (**a**) 2M ZnCl$_2$, (**b**) 2M ZnCl$_2$ + 10 wt.% PEG400, (**c**) 2M ZnSO$_4$, (**d**) 2M ZnSO$_4$ + 10 wt.% PEG400, varying the applied current density. Insets: magnified portion of each XRD diffraction patterns in the range 35–45°.

In general, SEM and XRD results suggest that the inkjet-printed Ti$_3$C$_2$T$_x$ coating seemed to not play any role in the preferential orientation alignment by epitaxial growth

of the zinc crystals, and the surface texture was solution-controlled rather than substrate-controlled. In fact, previous work demonstrated that the substrate zincophilicity could effectively regulate the Zn growth up to very narrow Zn thickness, and above a 1 nm-thick layer, the Zn electrolcrystallization was kinetically controlled rather than thermodynamically, so that the substrate affinity became negligible [56]. However, it is also likely that the very limited average lateral size of the $Ti_3C_2T_x$ 2D sheets used in our work (required for inkjet printing) negatively affected the epitaxial growth of the first Zn nuclei. In fact, the restacking of the small $Ti_3C_2T_x$ 2D sheets produced a huge density of $Ti_3C_2T_x$ edges and boundaries. As a consequence, this work may suggest that inkjet printing is not a convenient technique to prepare $Ti_3C_2T_x$ zincophilic coatings for [002]-textured Zn deposition in Zn-ion battery applications. Future works may finally address this question and will be devoted to the investigation of the effect of $Ti_3C_2T_x$ MXene sheet size on the texturing of plated Zn and its reversibility by multiple plating–stripping cycles.

## 4. Conclusions

This work focused on the Zn nucleation and growth behavior of inkjet-printed $Ti_3C_2T_x$ MXene substrates. Different electrolytes were investigated, namely zinc chloride- and sulfate-based solutions with and without PEG400 as an organic additive to improve the surface texture. We showed that, in chloride-based electrolytes, the fastest reaction rates were achieved. On the contrary, $SO_4^{2-}$ solutions were characterized by the highest nucleation overpotentials and charge transfer resistance as a consequence of the slower ion mobility in sulfate-based solutions or the adsorption of different anions at the $Ti_3C_2T_x$ surface. In agreement with recent literature, SEM and XRD analyses demonstrated that to achieve more [002]-textured Zn plating, excessively high reaction rates are unfavorable. In this regard, PEG400 effectively improved the preferential [002] orientation in both the $Cl^-$ and $SO_4^{2-}$ electrolytes, but, in the latter one, the excessive surface overpotential induced poor Zn deposition, favoring zinc hydroxide sulfate precipitation. This work also suggests that inkjet-printed $Ti_3C_2T_x$ MXene coatings are not effective in promoting the epitaxial growth of a Zn layer that is electrolyte-controlled.

**Supplementary Materials:** The following supporting information can be downloaded at: https://www.mdpi.com/article/10.3390/app14020682/s1, Figure S1: (a) SEM image of the multi-layered $Ti_3C_2T_x$ flakes after etching step and (b,c) the corresponding EDX quantiative analysis results; Figure S2: XRD pattern of the delaminated $Ti_3C_2T_x$ MXenes. For the analysis, a diluted delaminated $Ti_3C_2T_x$ aqueous suspension was vacuum filtered and dried to obtain a free-standing $Ti_3C_2T_x$ film; Figure S3: DLS of the $Ti_3C_2T_x$ ink employed for electrode MXene-coated electrode preparation; Figure S4: Deposited areal (dry) $Ti_3C_2T_x$ mass loaded as function of overprinting layers with (a) the fresh ink and (b) aged ink after 50 days of storage; Figure S5: EDX mapping of the 20 layer printed $Ti_3C_2T_x$ on SS316 disk substrate; Figure S6: (a) Picture of the SS316 disk coated with 20 layer of $Ti_3C_2T_x$ by inkjet printing. The typical bright bluish colour the $Ti_3C_2T_x$ MXene is visible. In (b) is reported the SEM of the cross section of the $Ti_3C_2T_x$ coating, showing a thickness of roughly 2 μm; Figure S7: CVs of $Ti_3C_2T_x$ electrode in different electrolytes, i.e., 2M $ZnCl_2$ (a), 2M $ZnCl_2$ + 10 wt.% PEG (b), 2M $ZnSO_4$ (c) and 2M $ZnSO_4$ + 10 wt.% PEG (d), recorded varying scan rate, i.e., 2, 5, 10, 25, 50, 100 mV s$^{-1}$; Figure S8: (a) picture of the Hull-cell employed in this work. The $Ti_3C_2T_x$ coated SS316 plate is placed on the oblique side of the trapezoidal cell while a Pt plate on the short side of the cell; (b) card for the local current density estimation with the cathode coupon underneath; Figure S9: SEM image of the pristine $Ti_3C_2T_x$ coated electrode.

**Author Contributions:** Conceptualization, E.G. and P.V.; methodology, P.V., V.M. and E.G.; software, P.V. and V.M.; validation, P.V. and V.M.; formal analysis, P.V., V.M. and E.G.; investigation, P.V. and V.M.; resources, V.M. and E.G.; data curation, V.M., P.V. and E.G.; writing—original draft preparation, P.V. and E.G.; writing—review and editing, P.V., L.M. and E.G.; supervision, L.M. and E.G. All authors have read and agreed to the published version of the manuscript.

**Funding:** This research received no external funding.

**Institutional Review Board Statement:** Not applicable.

**Informed Consent Statement:** Not applicable.

**Data Availability Statement:** Data are contained within the article.

**Conflicts of Interest:** The authors declare no conflicts of interest.

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
