# Peer review of "Zinc Plating on Inkjet-Printed Ti3C2Tx MXene: Effect of Electrolyte and PEG Additive"

_applsci, doi:10.3390/app14020682_

Round 1
Reviewer 1 Report
Comments and Suggestions for Authors
The authors’ article, “Zinc Plating on Inkjet Printed Ti3C2Tx MXene: Effect of Electrolyte and PEG400 Additive,” addresses a contemporary issue. However, in its current state, the article is not suitable for publication. The authors reference data in the supporting information, which is not included in the manuscript under review, making it challenging for the reviewer to predict its future content.
The process of preparing the MAX phase layer, where zinc is subsequently deposited, is only briefly described. The authors should provide a detailed description of the printed MAX phase layers, including their thickness and composition homogeneity, using techniques such as powder XRD, EDX, SEM, and AFM.
Questions arise such as: Is the homogeneity and thickness of the deposited MAX phase consistent after each print? What is the native print resolution? Could seed networks be formed by printing individual layers, where the epitaxial growth of deposited zinc crystals could occur preferentially?
The authors should also discuss the added value of depositing the MAX phase using the printing technique. They should evaluate the usability of the prepared electrodes in experimental batteries in terms of capacity and recyclability, specifically in the context of charge-discharge cycles. How many cycles can occur before the electrochemical degradation of the deposited layers begins?
Finally, the authors should compare their results with existing literature and discuss the advantages of their chosen approach. The authors are advised to revise their manuscript before resubmitting or submitting it elsewhere thoroughly.
Author Response
Comments
The authors’ article, “Zinc Plating on Inkjet Printed Ti3C2Tx MXene: Effect of Electrolyte and PEG400 Additive,” addresses a contemporary issue. However, in its current state, the article is not suitable for publication. The authors reference data in the supporting information, which is not included in the manuscript under review, making it challenging for the reviewer to predict its future content.
For the sake of clarity, we suppose that the reviewer only downloaded the .pdf file and not the .zip file that included both the main text and supplementary information file. We kindly ask reviewer to download the .zip file that include now the new version of the manuscript and supplementary information.
1) The process of preparing the MAX phase layer, where zinc is subsequently deposited, is only briefly described. The authors should provide a detailed description of the printed MAX phase layers, including their thickness and composition homogeneity, using techniques such as powder XRD, EDX, SEM, and AFM.
According to reviewer suggestion, description of the MXene printing process was improved in the experimental section, where the following sentence was included:
“A flat-bed thermal inkjet printer (Breva, IJet2L), equipped with Ti3C2Tx ink-filled HP45 cartridge, was employed for inkjet printing operations. Typically, the substrates were placed on the flat bed heated at 40 °C and 20 layers of the Ti3C2Tx ink were consecutively overprinted to achieve a consistent MXene load.”
Moreover, as suggested by the reviewer, SEM ,EDX, DLS and other information about printing reliability were added in the manuscript and the corresponding analyses graphs included in the SI. More specifically, the text was extended as follows:
“Ti3C2Tx MXenes were successfully obtained by the well-known procedure of mixed acid (HF-HCl) etching and subsequent delamination in LiCl solution. Successful etching was confirmed by SEM analyses (Figure S1a) that clearly showed the “open-book” morphology of the particles as result of the Al selective etching [33]. EDX analyses (Figure S1b,c) also reported a very low average Al to Ti atomic ratio (0.041), confirming successful Al removal. On the other hand, XRD (Figure S2) assessed the successful delamination to individual Ti3C2Tx MXenes sheets because of the left shift of the intense [002] peak to very low diffraction angle (6 °) as well as the progressive disappearance of the higher order peaks [45,46]. According to the average hydrodynamic diameter measured by DLS analysis (Figure S3), the average lateral size of the 2D MXene sheets was estimated to be 302 nm [41].
Inkjet printing is a well-known technique capable of quickly producing thin coatings on any substrate by the jetting of small droplets of a nanoparticle colloidal suspension though the nozzles of the printhead. Despite its wide industrial employment and suitability for the interfacial engineering regulation of the electrodes, just poor attention was devoted to inkjet printing and the usual slurry electrodes coating is usually preferred. The Ti3C2Tx MXenes ink we formulated was easily printable and Ti3C2Tx coated electrodes could be quickly produced. The consistency and repeatability of the printing process was checked my weighting the samples. Figure S4 shows that the even after more than 1 month of ink storage the printed Ti3C2Tx mass was highly reproducible, meaning that the as prepared ink was stable and no agglomeration or nozzle clogging occurred. Samples were prepared by printing 20 consecutive overlayers, resulting in a Ti3C2Tx mass load of 0.452 mg cm-2. The EDX mapping (Figure S5) of the bright bluish Ti3C2Tx coated electrodes revealed a very smooth surface and uniform elemental distribution. The thickness of the dried Ti3C2Tx 20 layer coating was estimated to be ~ 2 µm by the SEM analysis of the cross-section (Figure S6). ”
2) Is the homogeneity and thickness of the deposited MAX phase consistent after each print?
The homogeneity, reproducibility and thickness were checked by weighting the dried MXene printed mass. 20 overpriting layers consistently resulted in 0.452 mg/cm2 of printed MXenes, corresponding to a ~2 um thickness. More specifically, the following information was included in the manuscript:
“The consistency and repeatability of the printing process was checked my weighting the samples. Figure S4 shows that the even after more than 1 month of ink storage the printed Ti3C2Tx mass was highly reproducible, meaning that the as prepared ink was stable and no agglomeration or nozzle clogging occurred. Samples were prepared by printing 20 consecutive overlayers, resulting in a Ti3C2Tx mass load of 0.452 mg cm-2. The EDX mapping (Figure S5) of the bright bluish Ti3C2Tx coated electrodes revealed a very smooth surface and uniform elemental distribution.”
3) What is the native print resolution?
Native printer resolution was 600 dpi. The information was added in the experimental paragraph as follows:
“A flat-bed thermal inkjet printer (Breva, IJet2L), equipped with Ti3C2Tx ink-filled HP45 cartridge, was employed for inkjet printing operations. Native printer resolution was 600 dpi.”
4) Could seed networks be formed by printing individual layers, where the epitaxial growth of deposited zinc crystals could occur preferentially?
If we have understood correctly the reviewer comment, we agree that Ti3C2Tx MXene should effectively function as seed layer for epitaxial growth of Zn, as It was demonstrated by previous literature works. We supposed that the 2 µm-thick MXene printed coating could be effective in regulating the [002] texture of the growing zinc crystals. However, our findings suggests that the MXene layer was not effective alone in the proper Zn crystal alignment along the [002] orientation. On the other hand, the electrolyte played the big role as additive-free zinc sulfate produced much more [002] oriented and regular shaped zinc crystals respect to the zinc chloride electrolyte. These considerations are spread in the manuscript. For the sake of clarity we report here two examples:
“Comparing the PEG-free electrolytes, the electrolyte itself (specifically the counter ions Cl- or SO42-) seemed to play a major role in the growth of textured Zn coating rather than the Ti3C2Tx layer itself. In fact, the Ti3C2Tx coating should promote an epitaxial-like growth of the plated Zn layer because of the good matching between the two crystal lattices [37]. However, SEM analyses supported the electrochemical results suggesting that ions mobility and/or a different anion adsorption at the electrolyte/Ti3C2Tx interface mostly ruled the surface overpotential and the resulting Zn electro crystallization.”
“In general, SEM and XRD results suggested that the inkjet printed Ti3C2Tx coating seemed to not play any role in the preferential orientation alignment by epitaxial grow of the zinc crystals and the surface texture was solution-controlled rather than substrate controlled. In fact, previous work demonstrated that the substrate zincophilicity could effectively regulate the Zn growth up to very narrow Zn thickness and above 1 nm thick layer the Zn electrolcrystallization was kinetically controlled rather than thermodynamically, so that the substrate affinity became negligible [52]. However, it is also likely that the very limited average lateral size of the Ti3C2Tx 2D sheets used in our work (required for inkjet printing), negatively affected the epitaxial growth of the first Zn nuclei. In fact, the restacking of the small Ti3C2Tx 2D sheets produced a huge density of Ti3C2Tx edges and boundaries. As consequence, this work may suggest that inkjet printing is not a convenient technique to prepare Ti3C2Tx zincophilic coatings for [002]-textured Zn deposition in Zn-ion battery application.”
5) The authors should also discuss the added value of depositing the MAX phase using the printing technique.
We think that inkjet printing is a very suitable technique for surface modification of electrodes since thin and uniform coatings can be quickly produced by inkjet printing. On the other hand, the usual slurry coating is typically used to deposit thick layer of active material. In the framework of surface engineering, we think that poor attention was paid to electrode surface modification by inket printing. We thank the reviewer for the suggestion and, for the sake of clarity, we included the following consideration in the manuscript:
“Inkjet printing is a well-known technique capable of quickly producing thin coatings on any substrate by the jetting of small droplets of a nanoparticle colloidal suspension though the nozzles of the printhead. Despite its wide industrial employment and suitability for the interfacial engineering regulation of the electrodes, just poor attention was devoted to inkjet printing and the usual slurry electrodes coating is usually preferred. The Ti3C2Tx MXenes ink we formulated was easily printable and Ti3C2Tx coated electrodes could be quickly produced.”
6) They should evaluate the usability of the prepared electrodes in experimental batteries in terms of capacity and recyclability, specifically in the context of charge-discharge cycles.
The aim of the work was the investigation of the zinc plating behavior on the inkjet printed MXene electrode according to the electrolyte composition. We are aware that charge-discharge cycles testing of the electrodes could have made the work more valuable, however it was out of the aim of this work and it will be surely subject of future investigations. For the sake of clarity, few considerations about “future works and perspectives” were included in the manuscript as follows:
“Future works may finally address this question and will be devoted to the investigation of the effect of Ti3C2Tx MXene sheets size on the texturing of plated Zn and its reversibility by multiple plating-stripping cycles.”
7) How many cycles can occur before the electrochemical degradation of the deposited layers begins?
We thank the reviewer for the question, however this comment is strictly related to previous one and the reply to reviewer interrogative was included in previous reply.
8) Finally, the authors should compare their results with existing literature and discuss the advantages of their chosen approach.
To the best of our knowledge, no previous works can be found in literature about the investigation of Zn plating on inkjet-printed MXene electrodes. As we sparsely underlined in the manuscript, in qualitative terms our findings are coherent to the recent literatures when considering the surface overpotential effect on the Zn [002] texturing. Both the discussion of the electrochemical results and SEM analyses deeply focused on this consideration. In conclusive part of the Result paragraph we also highlighted that the MXene inkjet-printing modification approach is likely limited in the size of the MXene flakes that could override the epitaxial zn seed growth because of the huge density of MXene edges. However, as specified in relation to comment #6, this interrogative will be finally addressed in future work.
Reviewer 2 Report
Comments and Suggestions for Authors
The document “Zinc plating on inkjet printed Ti3C2Tx MXene: effect of electrolyte and PEG400 additive” focused on the study of electrodeposited Zn on printed Ti3C2Tx Mxene by changing electrolyte parameters. Electrochemical potentiostat tests were achieved under PEG400 to understand the Zn nucleation and growth.
Comments:
1) The abstract requires modification. Pay attention to the grammar structure because is not easy to follow
2) Add relevant information to the abstract section
3) The introduction section mentions some related works but it should be important to add others to clarify the importance and the difference of this manuscript in comparison with those previously published. Here there are some documents but authors can choose anything.
Toward a Practical Zn Powder Anode: Ti3C2Tx MXene as a Lattice-Match Electrons/Ions Redistributor | ACS Nano
Continuous Fabrication of Ti3C2Tx MXene-Based Braided Coaxial Zinc-Ion Hybrid Supercapacitors with Improved Performance | Nano-Micro Letters (springer.com)
In situ growth of ZnO nanosheets on Ti3C2Tx MXene for Superior-Performance Zinc-Nickel secondary battery - ScienceDirect
And clarify what is the novelty of this work in comparison with the last document (In situ growth of ZnO nanosheets on Ti3C2Tx MXene for Superior-Performance Zinc-Nickel secondary battery - ScienceDirect)
4) The methodology is well structured but it can be improved. Give more information about the electrochemical conditions.
5) Determine crystallite size, lattice parameter, and interplanar distance.
6) Add future work and perspective
Author Response
Comments and Suggestions for Authors
The document “Zinc plating on inkjet printed Ti3C2Tx MXene: effect of electrolyte and PEG400 additive” focused on the study of electrodeposited Zn on printed Ti3C2Tx Mxene by changing electrolyte parameters. Electrochemical potentiostat tests were achieved under PEG400 to understand the Zn nucleation and growth.
1) The abstract requires modification. Pay attention to the grammar structure because is not easy to follow
According to the reviewer suggestion, the abstract was overall revised, including relevant information of the main achievements of the work.
2) Add relevant information to the abstract section
Actions in respect to reviewer suggestions for improvements were addressed along with comment #1.
3) The introduction section mentions some related works but it should be important to add others to clarify the importance and the difference of this manuscript in comparison with those previously published. Here there are some documents but authors can choose anything:
- Toward a Practical Zn Powder Anode: Ti3C2Tx MXene as a Lattice-Match Electrons/Ions Redistributor | ACS Nano
According to reviewer suggestion, this reference was included in the introduction section (Ref #35)
- Continuous Fabrication of Ti3C2Tx MXene-Based Braided Coaxial Zinc-Ion Hybrid Supercapacitors with Improved Performance | Nano-Micro Letters (springer.com)
According to reviewer suggestion, the use of Ti3C2Tx MXene also for Zn-ion supercapacitor application was mentioned in the introduction section, along with the suggested reference work:
“Ti3C2Tx MXene can unlock a fast electrochemical kinetics in the plating/stripping process due to their high electronic conductivity and rapid Zn ions diffusion. In fact, Ti3C2Tx MXene have been employed also in Zn-ion supercapacitors owing to their outstanding Zn-ion adsorption capability [28,29].”
- In situ growth of ZnO nanosheets on Ti3C2Tx MXene for Superior-Performance Zinc-Nickel secondary battery - ScienceDirect
And clarify what is the novelty of this work in comparison with the last document (In situ growth of ZnO nanosheets on Ti3C2Tx MXene for Superior-Performance Zinc-Nickel secondary battery - ScienceDirect).
In this reference work, ZnO-coated Ti3C2 MXene particles are proposed as highly performing anodic material to improve plating-stripping of zinc in an alkaline Zinc-Nickel rechargeable battery. As consequence, both the material itself (the ZnO coating on MXene by simple chemical precipitation), the electrode preparation (conventional slurry formulation and coating on a brass net) and the electrolyte (satured ZnO in 6 M KOH) are completely different from our approach and we think they can hardly be compared in a meaningful way and the reference was not included in the revised manuscript. However, according to reviewer suggestion, the novelty of the work was better clarified in the introduction section:
“In this work, inkjet printing was employed as non-conventional technique, rarely reported in literature, for the surface modification of battery electrodes.”
4) The methodology is well structured but it can be improved. Give more information about the electrochemical conditions.
Detailed electrochemical conditions were reported in Experimental section and reminded in caption of related figures. However, according to reviewer suggestion some more clarifications were added in experimental section, namely:
“All the electrochemical test were performed withour stirring the solution. Tafel plots were built on the cathodic scan only of the 2 mV s-1 cyclic voltammetries. Electrochemical impedance spectroscopy (EIS) measurements were perfomed with the same three-electrode configuration described above…”
5) Determine crystallite size, lattice parameter, and interplanar distance.
According to reviewer suggestion, the crystallite size of the plated Zn at 5 mA cm-2 was calculated by means of Scherrer equation. The manuscript was integrated with this information as follows:
“In terms of crystallite dimensions, ZnCl2, with and without PEG400 and 2M ZnSO4 gave similar results, corresponding to average crystallites size of 48.8 nm, 57 nm and 48.8 nm respectively. Only in the case of 2M ZnSO4 + 10 wt.% PEG400 the calculated crystallites size was sensibly lower, counting to 37.92 nm coherently with the low efficiency of Zn deposition and lower grain growth rate.”
6) Add future work and perspective
According to the reviewer suggestion, a few considerations on future perspective were included in the manuscript as follows:
“Future works may finally address this question and will be devoted to the investigation of the effect of Ti3C2Tx MXene sheets size on the texturing of plated Zn and its reversibility by multiple plating-stripping cycles.”
Reviewer 3 Report
Comments and Suggestions for Authors
The present study examines Zn nucleation and growth behavior on Ti3C2Tx MXene substrates with two different electrolytes and with or without PEG400. The topic offers great potential in the field of energy storage. The results of this study are interesting, but additional electrochemical tests are urgently needed before they can be published in a scientific journal. The authors failed to present the cyclic and rate tests to demonstrate the effects of MXene and PEG400. I recommend a major revision with the following detailed comments:
1- The abstract should include a definition of PEG400. A more general term should also be used in the title.
2- The MXene material should be structurally characterized using various techniques, such as XRD, TEM, and XPS.
3- Symmetric and asymmetric battery tests should be conducted to demonstrate the effects of MXene and PEG400 on Coulombic efficiency, cyclic stability, and rate capability.
4- It is necessary to perform EIS tests following different cycles in order to determine whether MXene and PEG400 have any effects on kinetics during long-term cycling and compare resistance values over time.
5- Additional post-mortem characterizations are necessary to demonstrate the quality and reliability of materials/electrodes after long-term cycles.
Author Response
Comments
The present study examines Zn nucleation and growth behavior on Ti3C2Tx MXene substrates with two different electrolytes and with or without PEG400. The topic offers great potential in the field of energy storage. The results of this study are interesting, but additional electrochemical tests are urgently needed before they can be published in a scientific journal. The authors failed to present the cyclic and rate tests to demonstrate the effects of MXene and PEG400. I recommend a major revision with the following detailed comments:
1) The abstract should include a definition of PEG400. A more general term should also be used in the title.
According to the reviewer suggestions the title was modified as:
“Zinc plating on inkjet printed Ti3C2Tx MXene: effect of electrolyte and PEG additive”.
Moreover, a general definition of PEG was included in the abstract:
“Specifically, ZnCl2 and ZnSO4 solutions were employed, evaluating the effect of a relatively low molecular weight polyethylene-glycol (PEG400) addition to the electrolyte as additive”.
2) The MXene material should be structurally characterized using various techniques, such as XRD, TEM, and XPS.
According to reviewer suggestion, SEM ,EDX, DLS and other information about printing reliability were added in the manuscript and the corresponding analyses graphs included in the SI. More specifically, the text was extended as follows:
“Ti3C2Tx MXenes were successfully obtained by the well-known procedure of mixed acid (HF-HCl) etching and subsequent delamination in LiCl solution. Successful etching was confirmed by SEM analyses (Figure S1a) that clearly showed the “open-book” morphology of the particles as result of the Al selective etching. EDX (Figure S1b,c) also reported a very low average Al to Ti atomic ratio (0.041), confirming successful Al removal. On the other hand, XRD (Figure S2) assessed the successful delamination to individual Ti3C2Tx MXenes sheets because of the left shift of the intense [002] peak to very low diffraction angle (6 °) as well as the progressive disappearance of the higher order peaks. According to the average hydrodynamic diameter measured by DLS analysis (Figure S3), the average lateral size of the 2D MXene sheets was estimated to be 302 nm.
Inkjet printing is a well-known technique capable of quickly producing thin coatings on any substrate by the jetting of small droplets of a nanoparticle colloidal suspension though the nozzles of the printhead. Despite its wide industrial employment and suitability for the interfacial engineering regulation of the electrodes, just poor attention was devoted to inkjet printing and the usual slurry electrodes coating is usually preferred. The Ti3C2Tx MXenes ink we formulated was easily printable and Ti3C2Tx coated electrodes could be quickly produced. The consistency and repeatability of the printing process was checked my weighting the samples. Figure S4 shows that the even after more than 1 month of ink storage the printed Ti3C2Tx mass was highly reproducible, meaning that the as prepared ink was stable and no agglomeration or nozzle clogging occurred. Samples were prepared by printing 20 consecutive overlayers, resulting in a Ti3C2Tx mass load of 0.452 mg cm-2. The EDX mapping (Figure S5) of the bright bluish Ti3C2Tx coated electrodes revealed a very smooth surface and uniform elemental distribution. The thickness of the dried Ti3C2Tx 20 layer coating was estimated to be ~ 2 µm by the SEM analysis of the cross-section (Figure S6).
3) Symmetric and asymmetric battery tests should be conducted to demonstrate the effects of MXene and PEG400 on Coulombic efficiency, cyclic stability, and rate capability.
The aim of the work was the investigation of the zinc plating behavior on the inkjet printed MXene electrode according to the electrolyte composition. We are aware that charge-discharge cycles testing of the electrodes could have made the work more valuable, however it was out of the aim of this work and it will be surely subject of future investigations. For the sake of clarity, few considerations about “future works and perspectives” were included in the manuscript as follows:
“Future works may finally address this question and will be devoted to the investigation of the effect of Ti3C2Tx MXene sheets size on the texturing of plated Zn and its reversibility by multiple plating-stripping cycles.”
4) It is necessary to perform EIS tests following different cycles in order to determine whether MXene and PEG400 have any effects on kinetics during long-term cycling and compare resistance values over time.
We thank the reviewer for the question, however this comment is strictly related to previous one and the reply to reviewer interrogative was included in previous reply.
5) Additional post-mortem characterizations are necessary to demonstrate the quality and reliability of materials/electrodes after long-term cycles.
We thank the reviewer for the question, however this comment is strictly related to previous one and the reply to reviewer interrogative was included in reply to comment #3.
Reviewer 4 Report
Comments and Suggestions for Authors
I feel that this manuscript can be published in the esteemed journal Applied Sciences, ‘applies-2795693’ after the Minor Revision.
Comments-
1. The abstract should be eye-catching, and incorporate the novelty and findings of the work briefly.
2. Please check the writing very carefully, in some sentences the main subject is not clear.
3. The authors have missed the crucial point in discussing the importance of electrolytes and additives, please check.
4. Why too much change in overpotential 80 to 121, please explain properly
5. The Morphology analysis is not up to mark, please check and justify
6. Please check overall English, some of the English is not easy to understand as well as scientifically mismatched.

Comments on the Quality of English LanguageAuthor Response
Comments
1) The abstract should be eye-catching, and incorporate the novelty and findings of the work briefly.
According to the reviewer suggestion, the abstract was overall revised, including relevant information of the main achievements of the work.
2) Please check the writing very carefully, in some sentences the main subject is not clear.
According to reviewer suggestion an overall revision of the manuscript was done for improving. Refuses and grammar errors were checked within all manuscript.
3) The authors have missed the crucial point in discussing the importance of electrolytes and additives, please check.
The effect of the electrolyte composition on the zinc plating behavior was deeply discussed all over the manuscript and in particular in relation to the electrochemical results and SEM analyses, as well as in the conclusive part of the result paragraph. If the reviewer thinks it was not enough, we kindly ask the reviewer to more specifically point out what is lacking in the discussion of the results.
4) Why too much change in overpotential 80 to 121, please explain properly
We thank the reviewer for the comment. To properly explain this effect, the following consideration was added in the manuscript:
“In 2M ZnCl2 + 10 wt.% PEG400 the overpotential value just slightly increased from 62 mV to 71 mV, while in the case of 2M ZnSO4 + 10 wt.% PEG400 it consistently raised from 80 mV to 121 mV. This effect can be explained considering the different Zn-ion oligomer formation with PEG400 molecules. In fact, previous work demonstrated that the Zn2+-PEG oligomer strength strictly depends on the counter ions because of the different Zn ions solvation environment in zinc sulfate solutions respect to zinc halide (Cl, I, Br) ones [49]. Contrary to zinc halides electrolytes, in ZnSO4 solution zinc ions are strongly shielded by H2O molecules that firmly bind to PEG through H-bondings, resulting in stronger Zn-ion oligomer formation and, as consequence, overpotential increase.”
5) The Morphology analysis is not up to mark, please check and justify
According to reviewer suggestion, the analysis of SEM images was overall revised and few more comments were added in the manuscript:
“Individual Zn crystals seemed to grow vertically from the on Ti3C2Tx substrate, rather than forming a very compact and dense layer. A few of perfectly hexagonal Zn crystals were evidently growing from the surface of MXene coating (Figure 2i), with an approximate size of 100 µm”
“Zn crystals were scarcely visible and appeared in the whole current density range as irregular small particles with undefined morphology beneath and within the zinc hydroxide lamellas network”.
6) Please check overall English, some of the English is not easy to understand as well as scientifically mismatched.
According to reviewer suggestion an overall revision of the manuscript was done for improving, as in relation to comment #2. Refuses and grammar errors were checked within all manuscript.
Round 2
Reviewer 1 Report
Comments and Suggestions for Authors
I apologize in advance to the authors that I am not receiving the supplementary material in the first round. Please know that I emailed the editor to request one, but the assistant replied that the submitted manuscript contained no supplement.
I very much appreciate the care with which the authors have addressed all questions and comments. I believe that the changes made have improved the readability of the article. The paper, as now presented, may be of interest to the readers of Applied Sciences and, after minor corrections of typos, may be accepted for publication.
Just to give an example of a few errors found:
line 150 LDS (lithium dodecyl sulfate) - the abbreviation should be explained
line 202 checked my weighting should be checked by weighting
Figure S1 c) The table should contain the English term powder instead of an Italian polvere
Author Response
We thank the reviewer for the further suggestions.
All the corrections have been amended in the manuscript and SI.
Reviewer 2 Report
Comments and Suggestions for Authors
After reviewing the manuscript, it can be concluded that the present form of the document has the quality to be considered for publishing in the Applied Sciences Journal. Moreover, it should be mentioned that authors addressed and give response of question and suggestions.
Author Response
We thank the reviewer for the comment.
Reviewer 3 Report
Comments and Suggestions for Authors
Accept
Comments on the Quality of English LanguageMinor corrections are needed.
Author Response
We thank the reviewer for the comment. Few minor corrections have been amended in the manuscript and SI.